# ModuLoRA: Finetuning 2-Bit LLMs on Consumer GPUs by Integrating with Modular Quantizers

**Junjie Yin**                                                                    *jyin27@jhu.edu*
*Department of Computer Science*
*Johns Hopkins University*

**Jiahao Dong**                                                                   *jd787@cornell.edu*
*Department of Computer Science*
*Cornell University and Cornell Tech*

**Yingheng Wang**                                                                 *yw2349@cornell.edu*
*Department of Computer Science*
*Cornell University*

**Christopher De Sa**                                                             *cdesa@cs.cornell.edu*
*Department of Computer Science*
*Cornell University*

**Volodymyr Kuleshov**                                                            *kuleshov@cornell.edu*
*Department of Computer Science*
*Cornell University and Cornell Tech*

**Reviewed on OpenReview:** https://openreview.net/forum?id=r9p9CV52MV

## Abstract

We propose a memory-efficient finetuning algorithm for large language models (LLMs) that supports finetuning LLMs with 65B parameters in 2/3/4-bit precision on as little as one 24GB GPU. Our method, modular low-rank adaptation (MODULORA), integrates any user-specified weight quantizer with finetuning via low-rank adapters (LoRAs). Our approach relies on a simple quantization-agnostic backward pass that adaptively materializes low-precision LLM weights from a custom black-box quantization module. This approach enables finetuning 2-bit and 3-bit LLMs for the first time—leveraging state-of-the-art 2-bit QuIP# quantization and 3-bit OPTQ quantization—outperforming finetuning that relies on less sophisticated 4-bit and 8-bit methods. In our experiments, MODULORA attains competitive performance on text classification, natural language inference, and instruction following tasks using significantly less memory than existing approaches, and we also surpass the state-of-the-art ROUGE score on a popular summarization task. We release MODULORA together with a series of low-precision models as part of LLMTOOLS, a user-friendly library for quantizing, running, and finetuning LLMs on consumer GPUs.

## 1 Introduction

Large language models (LLMs) excel across diverse tasks such as code generation, instruction following, and reasoning (Brown et al., 2020; Scao et al., 2023; Zhang et al., 2022). However, the massive size of these models—often reaching into hundreds of billions of parameters—makes them challenging to deploy on downstream tasks and motivates research into efficient finetuning algorithms (Li & Liang, 2021; Hu et al., 2022).

Here, we propose modular low-rank adaptation (MODULORA), a memory-efficient finetuning algorithm for large language models (LLMs) that runs on consumer-grade hardware. For example, in 3-bit precision, MODULORA finetunes a LLaMA-30B model (Touvron et al., 2023) on one Nvidia RTX 3090 24GB GPU and a LLaMA-65B on one RTX A6000 48GB GPU. In 2-bit precision, MODULORA finetunes a LLaMA-30B or LLaMA-65B on one Nvidia RTX 3090 24GB GPU.

Our approach adds high-precision low-rank adapters to the low-precision 3-bit or 4-bit weights of a frozen base LLM obtained via modern quantization algorithms (Hubara et al., 2021; Yao et al., 2021; Frantar et al., 2023). Crucially, MODULORA does not specify its own quantization procedure—rather, it integrates with user-defined quantizers via a simple quantization-agnostic backward pass. This backward pass adaptively materializes low-precision LLM weights obtained from a black-box quantizer and integrates them with high-precision low-rank adapters.

We release MODULORA as part of LLMTOOLS, a user-friendly library that enables finetuning LLMs on consumer GPUs. When paired with the modern OPTQ quantizer (Frantar et al., 2023), MODULORA enables finetuning 3-bit LLMs for the first time, often outperforming methods based on less sophisticated 4-bit and 8-bit quantization. When paired with the state-of-the-art QuIP# quantizer Chee et al. (2023); Tseng et al. (2023), MODULORA enables finetuning 2-bit LLMs for the first time, matching methods' performance on less sophisticated 4-bit and 8-bit quantization method. Across tasks in classification, natural language inference, and instruction following, our low-precision models achieve competitive performance using significantly less memory than existing approaches. On a popular summarization benchmark, we attain a new state-of-the-art ROUGE score using a quantized LLaMA-65B model. We open-source all our low-precision models, including the first 3-bit family of Alpaca models that feature strong instruction-following performance at multiple model sizes. Our findings reveal that high performance can be achieved using smaller quantized LLMs than previously thought.

**Contributions.** In summary, this paper makes the following contributions: (1) we propose MODULORA, a memory-efficient finetuning method that operates over low-precision weights obtained via a user-specified black-box quantization module; (2) we release LLMTOOLS, a user-friendly Python library that features an implementation of MODULORA and that enables users to easily finetune the largest LLMs on consumer GPUs; (3) we provide empirical evidence that high performance on downstream tasks can be achieved with a smaller LLM than previously thought.

## 2 Background and Related Work

We are interested in finetuning a pre-trained LLM for downstream tasks (Li & Liang, 2021; Lester et al., 2021; Houlsby et al., 2019; Rebuffi et al., 2017). LLMs use a transformer architecture where almost all of the learnable weights—and almost all of the memory used to store these weights—appear in linear layers.[1] We let the weights and biases of these $n$ linear layers be denoted $\mathbf{W}^{(i)}$ and $\mathbf{b}^{(i)}$ for $i \in \{1, 2, ..., n\}$. Given a pretrained network, our goal is to finetune it for downstream tasks using much less working memory than would be needed to store all of the $\mathbf{W}$ in full precision.

### 2.1 Large Language Model Finetuning

Because of the high memory requirements needed to fine-tune and store all the weights of a LLM, practitioners have developed a variety of *parameter-efficient fine tuning* methods that learn in a lower dimensional space. These methods include tuning only the output layer (Devlin et al., 2018) and tuning the prompt or prefix passed as input to an LLM (Lester et al., 2021; Li & Liang, 2021; Liu et al., 2023a;b), as well as LoRA, which is the focus of this work.

**Low-Rank Adaptation (LoRA)** The LoRA algorithm (Hu et al., 2022) decomposes the weights $\mathbf{W}$ into[2] a sum of frozen base model weights $\mathbf{W}_0 \in \mathbb{R}^{d \times d}$ and a small additive low-rank adapter $\mathbf{AB}^\top$ consisting of

---

[1]These layers include the $K$, $V$, $Q$, and $O$ projection matrices of attention blocks and the linear layers of MLP blocks.
[2]For simplicity here we consider square weight matrices $\mathbf{W}$; the rectangular case is a straightforward generalization.

the product of two rectangular matrices $\mathbf{A}, \mathbf{B} \in \mathbb{R}^{d \times r}$, where $r > 0$ indicates the rank:

$$\mathbf{W} = \mathbf{W}_0 + \mathbf{A}\mathbf{B}^\top. \tag{1}$$

LoRA reduces the number of trained parameters by a factor of $2r/d$, lowering the storage, transmission, and task-switching overhead of inference on a system that already maintains the base model. However, LoRA must hold the base weights $\mathbf{W}_0$ in memory, which requires multiple high-end GPUs and precludes tuning large LLMs on commodity hardware.

## 2.2 Low-Precision Machine Learning

The computational requirements of modern machine learning models motivate a wide range of efficient machine learning algorithms (Li & Liang, 2021; Hu et al., 2022; Frantar et al., 2023).

**Quantization** Quantization methods for neural networks reduce the number of bits required to store model weights (Dong et al., 2019; 2020; Yao et al., 2022; Park et al., 2023). A $b$-bit quantization method has the form

$$(\hat{\mathbf{W}}_q, \mathbf{z}, \mathbf{s}) = \mathcal{Q}(\mathbf{W}) \qquad\qquad \hat{\mathbf{W}} = \mathcal{D}(\hat{\mathbf{W}}_q, \mathbf{z}, \mathbf{s}). \tag{2}$$

Here, the quantization algorithm $\mathcal{Q}$ takes a weight matrix $\mathbf{W} \in \mathbb{R}^{d \times d}$ (or its subset) and outputs a quantized version $\hat{\mathbf{W}}_q \in \{0, 1, \ldots, 2^{b-1}\}^{d \times d}$ (using $b$ bits to represent each entry of $\mathbf{W}$), as well as zero and scale parameters $\mathbf{z}, \mathbf{s} \in \mathbb{R}^d$ (in full precision). The dequantization algorithm $\mathcal{D}(\hat{\mathbf{W}}_q, \mathbf{z}, \mathbf{s})$ recovers an approximation $\hat{\mathbf{W}} \in \mathbb{R}^{d \times d}$ by rescaling the quantized weights as $\hat{\mathbf{W}} = \mathbf{s} \odot \hat{\mathbf{W}}_q + \mathbf{z}$, where $\odot$ denotes the Hadamard product, and $\odot, +$ are extended with numpy-style broadcasting.

Recently, Frantar et al. (2023) proposed OPTQ, a quantization algorithm that scales to modern LLMs. The method iteratively runs two steps over the weight columns: (1) quantize with nearest rounding and compute the error, (2) update the remaining weights with a scaled error. Many of our experiments finetune LLMs quantized with OPTQ.

Following OPTQ, Chee et al. (2023) proposed QuIP, a quantization algorithm that makes two-bit LLM compression viable for the first time. The method follows a 2-step procedure: (1) an adaptive rounding procedure that minimizes a quadratic proxy objective„ (2) an efficient pre- and post-processing procedure ensuring weight and Hessian incoherence through multiplication by random orthogonal matrices. Further, Tseng et al. (2023) proposed QuIP#, combining lattice codebooks with incoherence processing from QuIP to create state-of-the-art 2 bit quantized models. We show the performance of QuIP# (with $D_4$ codebooks) quantized LLMs on the SAMSum summarization experiment.

In concurrent work, Dettmers et al. (2023) proposed QLoRA, an approach for tuning quantized LLMs based on LoRA. While our work seeks to integrate with any user-defined quantization module (such as OPTQ), QLoRA defines its own quantization scheme, which is simpler than, say, OPTQ or QuIP. One advantage of our approach is support for 2-bit and 3-bit finetuning; QLoRA only supports 4-bit finetuning. We will also identify settings where using advanced quantizers yields performance gains over QLoRA. See Section 5.1 for details.

## 3 Low-Precision Low-Rank Adaptation with a Modular Quantizer

In this section, we describe modular low-rank adaptation (MODULoRA), a memory-efficient finetuning algorithm for large language models (LLMs) that leverages custom quantization algorithms and runs on consumer GPU hardware.

### 3.1 Low-Rank Adaptation of Low-Precision Models

The first step of our approach is *quantization*: we apply a black-box quantization algorithm $\mathcal{Q}$ to a set of pre-trained weight matrices $\mathbf{W}^{(i)}$. This yields quantized weights, zeros, and scales $(\hat{\mathbf{W}}_q^{(i)}, \mathbf{z}^{(i)}, \mathbf{s}^{(i)}) = \mathcal{Q}(\mathbf{W}^{(i)})$. We use $\hat{\mathbf{W}}_q^{(i)}$ to denote the quantized weights stored in low precision, while $\hat{\mathbf{W}}^{(i)}$ denotes the same weights materialized in high precision (both approximate the original weights $\mathbf{W}^{(i)}$). Crucially, we do not specify a quantization procedure $\mathcal{Q}$ as part of MODULORA—rather, we seek to support user-defined quantizers that are treated by our method is a black-box.

The core of our efforts focuses on *finetuning* the base quantized model. Our method first modifies the network by replacing each linear layer—originally defined by the affine map $x \mapsto x(\mathbf{W}^{(i)})^\top + \mathbf{b}^{(i)}$—with the reparameterized low precision ModuLoRALinear layer in Figure 1, given by

$$x \mapsto x(\hat{\mathbf{W}}^{(i)})^\top + x\mathbf{B}^{(i)}(\mathbf{A}^{(i)})^\top + \mathbf{b}^{(i)}. \quad (3)$$

Here $\mathbf{A}^{(i)}, \mathbf{B}^{(i)} \in \mathbb{R}^{d \times r}$ are learnable parameters initialized as in Hu et al. (2022), and $\hat{\mathbf{W}}^{(i)} = \mathcal{D}(\hat{\mathbf{W}}_q^{(i)}, \mathbf{z}^{(i)}, \mathbf{s}^{(i)})$ is the fixed dequantized weight matrix. Note that this is algebraically (but not computationally) equivalent to transforming the quantized matrix as given in (1). Lastly, MODULORA fits the $\mathbf{A}^{(i)}$ and $\mathbf{B}^{(i)}$ using backprop and gradient-based learning.

```python
class ModuLoRALinear(Module):
    """Linear ModuLoRA Layer"""
    def __init__(self, ...):
        self.hatWq_z_s = quantize(pretrained_W)
        (self.A, self.B) = lora_init(...)
    def forward(self, x):
        (hatWq, z, s) = self.hatWq_z_s
        return LPLinear.apply(x, hatWq, z, s) \
            + (x @ self.B) @ self.A.t() + self.bias

class LPLinear(Function):
    """Low-Precision Linear Map"""
    @staticmethod
    def forward(ctx, input, hatWq, z, s):
        ctx.save_for_backward(hatWq, z, s)
        hatW = dequantize(hatWq, z, s)
        output = input @ hatW.t()
        return output # hatW is deallocated
    @staticmethod
    def backward(ctx, grad_output):
        hatWq, z, s = ctx.saved_tensors
        # we recompute hatW
        hatW = dequantize(hatWq, z, s)
        grad_input = grad_output @ hatW
        # here hatW can be deallocated
        return grad_input, None, None, None
```

Figure 1: PyTorch pseudocode for MODULORA.

A key challenge in this procedure is to efficiently perform computations with high-precision and low-precision tensors. Clearly, the forward pass requires multiplying by weights stored in quantized $\hat{\mathbf{W}}_q^{(i)}$'s. Below, we derive the backward pass for $\mathbf{A}^{(i)}, \mathbf{B}^{(i)}$ and show that it also requires multiplying by the transpose of the $\hat{\mathbf{W}}_q^{(i)}$'s.

### 3.1.1 The Structure of a Quantized Backward Pass

We illustrate the technical challenges that arise in the design of a quantized backward pass in the context of a network of $n$ ModuLoRALinear layers. Each ModuLoRALinear is effectively a fully connected layer with reparameterized dense weights defined as

$$\mathbf{W}_l^{(i)} = \hat{\mathbf{W}}^{(i)} + \mathbf{A}^{(i)}(\mathbf{B}^{(i)})^\top, \quad (4)$$

biases $\mathbf{b}^{(i)}$, and outputs $\mathbf{y}_i$ for $i = 1, 2, ..., n$. We use $\bar{\mathbf{y}}_i = \mathbf{W}_l^{(i)}\mathbf{x} + \mathbf{b}^{(i)}$ to denote the pre-activation output of the $i$-th step and we use $L$ to denote the loss. The backward pass seeks to compute gradients $\mathrm{d}L/\mathrm{d}\mathbf{A}^{(i)}$ and $\mathrm{d}L/\mathrm{d}\mathbf{B}^{(i)}$, where we overload the Leibniz notation for derivatives to also denote gradients. By the chain rule,

$$\frac{\mathrm{d}L}{\mathrm{d}\mathbf{A}^{(i)}} = \frac{\mathrm{d}L}{\mathrm{d}\bar{\mathbf{y}}_i} \cdot \frac{\mathrm{d}\bar{\mathbf{y}}_i}{\mathrm{d}\mathbf{A}^{(i)}}. \quad (5)$$

Because of the additive structure of the weights $\mathbf{W}_l^{(i)}$ in (4), $\mathrm{d}\mathbf{y}_i/\mathrm{d}\mathbf{A}^{(i)}$ is straightforward to handle as it is not a function of the quantized weights $\hat{\mathbf{W}}_q^{(i)}$. The second term can be computed via the chain rule of calculus as

$$\frac{\mathrm{d}L}{\mathrm{d}\bar{\mathbf{y}}_i} = \frac{\mathrm{d}L}{\mathrm{d}\bar{\mathbf{y}}_{i+1}} \cdot \frac{\mathrm{d}\bar{\mathbf{y}}_{i+1}}{\mathrm{d}\mathbf{y}_i} \cdot \frac{\mathrm{d}\mathbf{y}_i}{\mathrm{d}\bar{\mathbf{y}}_i}, \quad (6)$$

where $\mathrm{d}\mathbf{y}_i/\mathrm{d}\bar{\mathbf{y}}_i$ is the derivative of the activation function, and $\mathrm{d}\bar{\mathbf{y}}_{i+1}/\mathrm{d}\mathbf{y}_i = (\mathbf{W}_l^{(i)})^\top = (\hat{\mathbf{W}}^{(i)})^\top + \mathbf{B}^{(i)}(\mathbf{A}^{(i)})^\top$.

The above derivations indicate that computing the gradient $\mathrm{d}L/\mathrm{d}\mathbf{A}^{(i)}$ (the argument for $\mathrm{d}L/\mathrm{d}\mathbf{B}^{(i)}$ is identical) requires performing a matrix-vector multiply $\frac{\mathrm{d}L}{\mathrm{d}\mathbf{y}_{i+1}} \cdot (\hat{\mathbf{W}}^{(i)})^\top$ between a high-precision vector $\frac{\mathrm{d}L}{\mathrm{d}\mathbf{y}_{i+1}}$ with a quantized matrix $(\hat{\mathbf{W}}^{(i)})^\top$. Performing this multiplication in a stable and efficient way is a challenge that we must address.

### 3.1.2 Efficient Mixed-Precision Computation of Forward and Backward Passes

If we could precompute all dequantized weight matrices $(\hat{\mathbf{W}}^{(i)})^\top$ in a high-precision format, our challenge would be solved: the matrix-vetor multiplication $\frac{\mathrm{d}L}{\mathrm{d}\mathbf{y}_{i+1}} \cdot (\hat{\mathbf{W}}^{(i)})^\top$ in the backward pass would operate over two high-precision arrays, and would not introduce questions of efficiency and stability.

Unfortunately, precomputing all dequantized weight matrices $(\hat{\mathbf{W}}^{(i)})^\top$ requires the same amount of GPU memory as it would take to store the original high-precision LLM. For this computation to fit on consumer GPU hardware, we need to avoid manifesting all the $\hat{\mathbf{W}}^{(i)}$ in memory at once. Using (3) naively, backprop would store all the $\hat{\mathbf{W}}^{(i)}$ from the forward pass to use them in the backward pass.

**Efficient Mixed Precision Computation.** Our strategy is to *recompute* the high-precision materialization $\hat{\mathbf{W}}^{(i)}$ of the quantized $\hat{\mathbf{W}}_q^{(i)}$ in the backward pass rather than save it (Figure 1). In the `LPLinear` function, the `forward` method dequantizes $\hat{\mathbf{W}}^{(i)}$ and performs multiplication. Similarly, `backward` re-dequantizes $\hat{\mathbf{W}}^{(i)}$ and computes the gradient derived in Appendix **??** via dynamic programming. The `hatW` goes out of scope and can be freed at the end of each method, so only one $\hat{\mathbf{W}}^{(i)}$ is ever stored in memory at any given time.

The amount of memory used in the forward pass of the `LPLoRA` module is small: all the intermediates are either the same size as the input $x$, or even smaller (e.g. if $x \in \mathbb{R}^{m \times d}$ then `x @ self.B` is of size $\mathbb{R}^{m \times r}$ for $r \ll d$). The amount of additional computation involved is also small: the dequantization procedure $\hat{\mathbf{W}} = \mathbf{s} \odot \hat{\mathbf{W}}_q + \mathbf{z}$ only requires multiplying and adding a scalar to each row of $\hat{\mathbf{W}}_q$.

**Increasing Efficiency Further.** Figure 1 depicts a *weight materialization* strategy in which $\hat{\mathbf{W}}^{(i)}$ is fully materialized at each layer in both forward and backward passes. To further reduce memory, we can materialize elements of $\hat{\mathbf{W}}^{(i)}$ only as needed. For many quantization algorithms (Nagel et al., 2020; Frantar et al., 2023), we can perform *row materialization*: dequantize $\hat{\mathbf{W}}^{(i)}$ one row at a time and immediately multiply it with an input $\mathbf{x}$. MODULORA also naturally generalizes to any direct vector-by-quantized-matrix product *subroutine provided by the quantizer $\mathcal{Q}$*, in which case materializing any part of $\hat{\mathbf{W}}^{(i)}$ may be unnecessary.

### 3.2 LLMTools: A Library for Efficient LLM Finetuning Using ModuLoRA.

We implement MODULORA as part of LLMTOOLS, a user friendly library that enables users to interact with the largest LLMs on consumer hardware. The LLMTOOLS library enables finetuning LLMs in 2-bit, 3-bit, and 4-bit precision using the MODULORA algorithm. It also provides an easy-to-use Python API for quantization, inference, and finetuning, as well as modular support for multiple quantizers, LLMs (including LLaMA1, LLaMA2, BLOOM, and OPT), and optimization algorithms (including all that are compatible with the Hugging Face Trainer class). Lastly, LLMTOOLS supports easily loading datasets and sharing models via the HuggingFace Hub. Our code is available at: `https://github.com/kuleshov-group/llmtools`; our evaluation code to reproduce our results is available at: `https://github.com/kuleshov-group/MODULoRA-Experiment`.

A key quantization algorithm implemented in LLMTOOLS is OPTQ (Frantar et al., 2023). In order to integrate OPTQ with LoRA-based finetuning, LLMTOOLS provides efficient CUDA implementations of mixed-precision matrix-vector multiplication, including row and weight materialization. We provide CUDA kernels for both row and weight materialization in both the forward and backward passes. For maximum efficiency, we materialize elements of $\hat{\mathbf{W}}_q^{(i)}$ in float16. The base quantized LLM models are represented via weights $\hat{\mathbf{W}}_q^{(i)}$ stored in 3 or 4 bits, with scales and zeros $\mathbf{s}^{(i)}, \mathbf{z}^{(i)}$ as well as biases $\mathbf{b}^{(i)}$ all stored as float16. Similarly, to integrate QuIP# with LoRA, LLMTOOLS provides CUDA kernels for weight re-materialization and orthogonal matrices multiplication in the forward and backward passses. The base quantized LLM models are represented via weights $\hat{\mathbf{W}}_q^{(i)}$ stored in 2 bits.

## 4 Experiments

### 4.1 Setup

**Models.**   We evaluate MODULORA and LLMTOOLS on the recent LLaMA (Touvron et al., 2023) family of models, as well as open-source BLOOM (Scao et al., 2023) and OPT models (Zhang et al., 2022). We quantize the models to 3 bits and 4 bits using OPTQ as in Frantar et al. (2023) with calibration 128 samples from C4 (Raffel et al., 2020). We quantize the models to 2 bits using QuIP# as in Chee et al. (2023); Tseng et al. (2023) with $D_4$ lattice codebooks[3].

**Baseline.**   We use LoRA (as implemented in the PEFT library (Mangrulkar et al., 2022)) to finetune models quantized in 8 bits using the BitsAndBytes library (Dettmers et al., 2022); we also compare to full-precision results from the literature. In concurrent work, Dettmers et al. (2023) proposed QLoRA, a related 4-bit finetuning algorithm implemented in the BitsAndBytes library. Accordingly, we present an experimental comparison of QLoRA with our approach, along with an in-depth discussion.

**Training.**   We finetune all models on NVIDIA TITAN, 3090, and A6000 GPUs (depending on the model) with a LoRA rank of $r = 8$ and alpha of $a = 32$, and report results from 3 random seeds. We set up the training procedure following Hu et al. (2022), with slight variation to accommodate our particular language models. For a fair comparison with the concurrent work by Dettmers et al. (2023), we use the exact same hyperparameter set up. Please see Appendix C for details on the hyperparameters used for each of our experiment.

### 4.2 Text Classification

**Data & Metrics.**   We start with a simple text classification task where we seek to classify a short text snippet (up to 50 words) into its genre (e.g., fiction, telephone chat, etc.). We finetune 13B to 65B LLaMA models on 392,702 snippets from five genres and evaluate on 9,815 held out instances (Williams et al., 2018), reporting accuracy. This yields a challenging classification task for LLMs of all sizes.

| LLAMA Tuning | Quantizer | 13B | 30B | 65B |
|---|---|---|---|---|
| LLMTOOLS (3-bit) | OPTQ | 93.5 $\pm$ 0.7 | 97.0 $\pm$ 0.9 | 97.2 $\pm$ 0.8 |
| LLMTOOLS (4-bit) | OPTQ | 92.9 $\pm$ 0.7 | 96.3 $\pm$ 1.0 | 98.0 $\pm$ 0.9 |
| Bits&Bytes (8-bit) | LLM.int8() | 93.0 $\pm$ 0.7 | 93.7 $\pm$ 1.0 | 98.6 $\pm$ 1.0 |

Table 1: Text classification accuracy (%) for LLAMAs finetuned with LoRA & MODULORA in 3, 4, 8 bits.

**Results.**   We observe that classification accuracy consistently improves as we increase the number of parameters of the LLM. MODULORA combined with a 3-bit or a 4-bit LLM offers comparable performance to 8-bit finetuning in Bits&Bytes while using significantly less memory (Table 1).

### 4.3 Natural Language Inference

**Data & Metrics.**   Next, we finetune LLMs on natural language inference tasks. The model is asked to predict a label from a small set (entailment, contradiction, or neutral) after being presented with a sentence pairing (a hypothesis and premise sentence pair). We finetune 7B to 65B LLaMA models on the Multi-Genre Natural Language Inference Corpus (MNLI)  (Williams et al., 2018) and evaluate on the matched test sets (in-domain examples), reporting accuracy. Baselines from GPT-3 and T5 are included, as presented in Hu et al. (2022) and  Chung et al. (2022).

**Results.**   Our 2-bit and 3-bit 65B LLaMA model matches the performance of a full-precision GPT-3+LoRA baseline. We also find that **3-bit and 4-bit models from LLMTools outperform 8-bit models from the Bits&Bytes library for the entire model size range**. 2-bit, 3-bit and 4-bit MODULORA models

---

[3]QuIP# also introduces $E_8$ based codebooks, which achieve an even lower element-wise mean squared error (MSE) than $D_4$ codebooks. With the application of $E_8$ codebooks, 2-bit QuIP# finetuning could potentially yield stronger results.

either match or outperform their 4-bit QLoRA counterparts, often using less memory because of lower precision models.

*Baselines*

| Models | Finetuning Adaptation | Model Size | # Trainable Parameters | MNLI-m (*accuracy*) |
|---|---|---|---|---|
| GPT-3 | Full Finetuning | 175B | 175,255.8M | $89.5 \pm 0.1$ |
| GPT-3 | Adapter | 175B | 40.1M | $91.5 \pm 0.1$ |
| GPT-3 | LoRA | 175B | 4.7M | $91.7 \pm 0.1$ |
| T5 | Full Finetuning | 11B | 11,307.4M | $\mathbf{92.2} \pm 0.1$ |

| LLaMA Finetuning | Quantizer | 7B | 13B | 30B | 65B |
|---|---|---|---|---|---|
| LLMTools (2-bit) | QuIP#($D_4$) | $86.59 \pm 0.5$ | $87.42 \pm 0.5$ | $89.72 \pm 0.5$ | $90.85 \pm 0.5$ |
| LLMTools (3-bit) | OPTQ | $88.98 \pm 0.2$ | $90.20 \pm 0.2$ | $91.09 \pm 0.2$ | $91.42 \pm 0.1$ |
| LLMTools (4-bit) | OPTQ | $89.31 \pm 0.2$ | $90.41 \pm 0.2$ | $91.31 \pm 0.1$ | $\mathbf{91.59} \pm 0.2$ |
| Bits&Bytes (4-bit) | QLoRA | $89.28 \pm 0.2$ | $89.67 \pm 0.2$ | $91.22 \pm 0.1$ | $91.36 \pm 0.2$ |
| Bits&Bytes (8-bit) | LLM.int8() | $88.95 \pm 0.1$ | $90.08 \pm 0.1$ | $91.15 \pm 0.1$ | $91.55 \pm 0.1$ |

Table 2: Natural language inference on the MNLI-m dataset evaluated using classification accuracy (%). Our LLaMA-65B-3bit model approaches state-of-the-art scores using significantly less memory.

## 4.4 Abstractive Summarization

**Data & Metrics.** We finetune 7B-65B LLaMA and 7B-13B OPT models on the SAMSum dataset (Gliwa et al., 2019), consisting of 14,732 (text, summary) training pairs and 819 test pairs. Our methodology fully mirrors the evaluation of GPT-style models finetuned using LoRA (Hu et al., 2022). We evaluate summarization quality using ROUGE-1/2/L; we include GPT-3 baselines from Hu et al. (2022).

**Results.** Our 4-bit 65B LLaMA models finetuned with MODULORA outperform the GPT-3 baseline and even **reach new state-of-the-art performance** on this dataset (Table 3). Importantly, MODU-LORA demonstrates performance improvements over the 4-bit QLoRA and the 8-bit BitsAndBytes methods. In the 7B to 65B model size range, MODULORA models (3-bit or 4-bit) outperform 8-bit LoRAs in BitsAndBytes and LLM.int8() and 4-bit LoRAs in BitsAndBytes and QLoRA. MODULORA models (2-bit) match the performance of 8-bit LoRAs in BitsAndBytes and LLM.int8() and 4-bit LoRAs in BitsAndBytes and QLoRA. We argue that a data-driven lower precision quantization scheme can improve over a higher precision zero-shot quantizer like LLM.int8(). Switching from 4-bit to 3-bit, and then from 3-bit to 2-bit, precision within MODULORA reduces ROUGE by only about 1%.

*Baselines*

| Models | Finetuning Adaptation | # Trainable Parameters | SAMSum (*Rouge 1/2/L*) |
|---|---|---|---|
| GPT-3 | Full Finetuning | 175,255.8M | 52.0 / 28.0 / 44.5 |
| GPT-3 | Adapter | 40.1M | 53.2 / 29.0 / 45.1 |
| GPT-3 | LoRA | 4.7M | 53.8 / 29.8 / 45.9 |
| Pegasus | SliC | 2B | **54.4 / 29.9 / 45.9** |

| LLAMA Finetuning | Quantizer | 7B | 13B | 30B | 65B |
|---|---|---|---|---|---|
| LLMTools (2-bit) | QuIP# ($D_4$) | 49.2 / 26.9 / 42.7 | 50.7 / 28.6 / 44.4 | 51.6 / 30.2 / 46.4 | 52.3 / 30.5 / 46.8 |
| LLMTools (3-bit) | OPTQ | 51.2 / 28.2 / 44.0 | 52.4 / 29.6 / 45.1 | 53.6 / 30.8 / 46.3 | 54.1 / 30.9 / 46.5 |
| LLMTools (4-bit) | OPTQ | 51.7 / 28.3 / 44.4 | 53.2 / 30.2 / 46.1 | 53.9 / 31.2 / 46.9 | **55.9 / 32.7 / 49.0** |
| Bits&Bytes (4-bit) | QLoRA | 51.6 / 28.3 / 44.5 | 51.3 / 28.1 / 44.1 | 53.0 / 30.2 / 45.7 | 53.8 / 30.5 / 45.9 |
| Bits&Bytes (8-bit) | LLM.int8() | 51.9 / 28.1 / 44.5 | 51.3 / 28.2 / 43.6 | 50.8 / 28.4 / 44.1 | 53.9 / 30.4 / 46.3 |

Table 3: Abstractive summarization on the SAMSum dataset evaluated using ROUGE 1/2/L. Our LLAMA-65B-4bit model obtains state-of-the-art ROUGE scores. All metrics have $\pm 0.5$ confidence intervals.

| SAMSum Performance | Quantizer | 7B | 13B |
|---|---|---|---|
| LLMTOOLS (3-bit) | OPTQ | 51.2 / 28.2 / 44.0 / 44.2 | 52.4 / 29.6 / 45.1 / 45.1 |
| | RTN | 50.7 / 27.2 / 43.6 / 43.6 | 51.1 / 28.7 / 44.3 / 44.5 |
| LLMTOOLS (4-bit) | OPTQ | 51.7 / 28.3 / 44.4 / 44.4 | 53.2 / 30.2 / 46.1 / 46.1 |
| | RTN | 51.2 / 28.5 / 44.2 / 44.2 | 52.5 / 29.9 / 45.5 / 45.5 |

Table 4: OPTQ and RTN quantization with different LLaMA model sizes on the SAMSum dataset. The evaluation was done on ROUGE 1/2/L/LSum.

**Round-to-Nearest Quantization** We also perform an ablation where we replace the OPTQ quantizer with a rount-to-nearest (RTN) approach (Table 4); OPTQ performs better than RTN, highlighting the importance of advanced quantizers.

**Other Model Families** We also apply LLMTOOLS to the OPT (Zhang et al., 2022) families of models (Table 5). Although these models perform worse than LLaMA, MODULORA matches or outperforms more memory-intensive 4-bit and 8-bit finetuning, which is consistent with our results on LLaMA.

| OPT Finetuning | Quantizer | 13B | 30B |
|---|---|---|---|
| LLMTOOLS (3-bit) | OPTQ | 48.8 / 26.7 / 41.9 | **49.9 / 27.1 / 42.5** |
| LLMTOOLS (4-bit) | OPTQ | 49.3 / 26.8 / 42.0 | 49.6 / 27.1 / 42.4 |
| Bits&Bytes (4-bit) | QLoRA | 49.2 / 27.0 / 42.1 | 49.9 / 27.0 / 42.5 |
| Bits&Bytes (8-bit) | LLM.int8() | 48.8 / 26.5 / 41.7 | 49.3 / 27.1 / 42.3 |

Table 5: Abstractive summarization with OPT models on the SAMSum dataset. MODULORA in 3-bit and 4-bit precision matches ROUGE 1/2/L scores of 4-bit and 8-bit baselines. All metrics have $\pm 0.5$ confidence intervals.

### 4.5 Instruction Following

**Data & Metrics.** We finetune 7B-65B LLaMA models on the Alpaca dataset (Taori et al., 2023), consisting 52,000 instructions, as well on the CodaAlpaca dataset (Chaudhary, 2023), consisting of 20K code generation instructions (ses Table 9). We evaluate our Alpaca instruction-tuned models on the BigBenchHard (BBH) benchmark (Suzgun et al., 2022), consisting of 23 challenging tasks on which LLMs do not exceed human performance. We evaluate 3-shot performance via "answer-only" prompting and use exact match accuracy as our measurement standard, testing on 6,511 samples ($\sim 1.5$k tokens each). We include Flan and LLaMA baselines from Chia et al. (2023).

**Results.** We find that 3-bit and 4-bit performance drops only slightly relative to 8-bit and 16-bit. 2-bit models, despite their aggressive compression, match the performance of 4-bit QLoRA in smaller model sizes. Crucially, **4-bit and 3-bit 65B models outperform 8-bit and 16-bit 30B models**, despite using fewer total bits. Furthermore, 4-bit MODULORA compares well to 4-bit QLoRA, and provides consistent performance improvements, especially at smaller model sizes, where sophisticated quantization ought to provide greater benefits. This further highlights the benefits of one-shot quantization methods. Appendix B also reports experiments on the CodeAlpaca dataset.

### 4.6 Memory Requirements

We show the memory required to perform finetuning on MNLI-M for different LLaMA model sizes in table 7. MODULORA significantly minimizes the memory requirements for finetuning on these models. We plot the memory requirements in figure 2 for better visualization. As the model size increases to 65B, MODULORA uses only about 6% of the memory to run memory-efficient finetuning method LoRA. As the

*Baselines*

| Model | Method | Quantizer | BASE (250M) | L (780M) | XL (3B) | XXL (11B) |
|---|---|---|---|---|---|---|
| FLAN-T5 | No Finetuning | None | 30.8 | 30.3 | 39.9 | 47.4 |

| Model | Methods | Quantizer | 7B | 13B | 30B | 65B |
|---|---|---|---|---|---|---|
| LLaMA | LLMTools (2-bit) | QuIP# ($D_4$) | $30.3 \pm 0.7$ | $33.3 \pm 0.6$ | $37.0 \pm 0.9$ | $39.3 \pm 0.9$ |
| | LLMTools (3-bit) | OPTQ | $31.1 \pm 0.4$ | $35.3 \pm 0.2$ | $37.2 \pm 0.6$ | $43.3 \pm 0.4$ |
| | LLMTools (4-bit) | OPTQ | $33.1 \pm 0.2$ | $36.2 \pm 0.4$ | $40.4 \pm 0.2$ | $43.7 \pm 0.4$ |
| | Bits&Bytes (4-bit) | QLoRA | $31.9 \pm 0.1$ | $35.4 \pm 0.2$ | $39.0 \pm 0.4$ | $43.5 \pm 0.5$ |
| | Bits&Bytes (8-bit) | LLM.int8() | $33.3 \pm 0.3$ | $36.8 \pm 0.2$ | $39.1 \pm 0.5$ | $44.7 \pm 0.4$ |
| | No Finetuning | None | 30.9 | 37.1 | 39.3 | 42.6 |

Table 6: Instruction-tuned models evaluated on BigBench Hard (BBH). We finetune LLaMA models on the Alpaca dataset in 2 to 16 bits. We provide exact standard deviation here.

table and figure illustrates, with MODULORA it's possible to not only run inference but also finetune 65B model on a single 24GB GPU. To produce this table, we run our quantizer-agnostic forward/backward passes for the entire LLaMA model size range with batch size 1 and maximum sequence length 128 on MNLI-m.

| LLaMA Finetuning | 7B | 13B | 30B | 65B |
|---|---|---|---|---|
| LLMTools (2-bit) | 3.2 GB | 5.4 GB | 11.4 GB | 21.8 GB |
| QLoRA (4-bit) | 5.2 GB | 8.6 GB | 19.5 GB | 36.7 GB |
| Full Precision (LoRA) | 38.4 GB | 73.9 GB | 183.3 GB | 360.4 GB |

Table 7: Memory requirements to finetune LLaMA models on MNLI-M with batch size 1 and maximum sequence length 128. For comparison, we include the memory requirements to finetune on LoRA and QLoRA.

# 5 Discussion

## 5.1 Comparison to Related Work

**Comparison to QLoRA** In concurrent work, Dettmers et al. (2023) proposed QLoRA, a related approach for finetuning a quantized LLM. We highlight methodological and experimental differences below. From a methods perspective, MODULORA integrates with a user-specified black-box quantization module. In our experiments, we find that using a sophisticated data-driven quantizer like OPTQ improves performance over simpler zero-shot strategies, e.g., a round-to-nearest baseline. Unlike MODULORA, QLoRA defines a quantization approach similar to RTN, but also introduces a specialized packing routine, quantization of zeros and scales, and other innovations.

From an experiments and capabilities perspective, integrating with OPTQ enables MODULORA to fintune models quantized in 2-bits and 3-bits, which QLoRA cannot do. Lastly, we identify settings where MODULORA yields LLMs with better performance than LLMs from QLoRA; this gap is likely due to the use of improved quantizers.

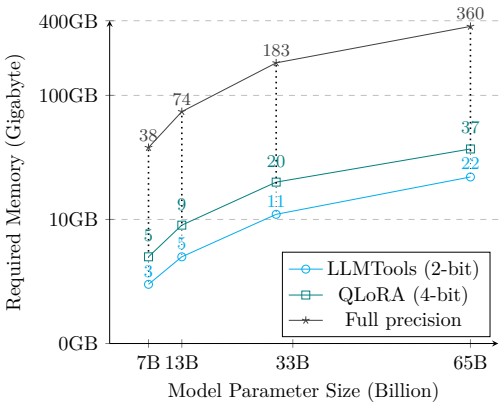

Figure 2: Visualization of memory requirements for different LLaMA model sizes with different methods.

**Comparison to Other Parameter-Efficient Finetuning Methods**   Recent Parameter-Efficient Fine-tuning (PEFT) methods have encompassed a range of techniques such as prompt tuning (Lester et al., 2021; Li & Liang, 2021; Qin & Eisner, 2021; Liu et al., 2022b), modification of the embedding layer inputs (An et al., 2022) or hidden states (Liu et al., 2022a), inclusion of full layers (Houlsby et al., 2019), only tuning biases (Zaken et al., 2021), and others (Sung et al., 2021; Karimi Mahabadi et al., 2021). An important shortcoming of these methods is the need to store in memory a significant amount of frozen base model parameters. This limits their ability to finetune the largest LLMs on consumer GPU, a limitation that we address.

## 5.2   Running LLMs on Consumer GPUs

**Efficient LLM Algorithms**   The computational requirements of modern deep neural networks motivate a wide range of efficient machine learning algorithms. Quantization methods reduce the number of bits required to store weights (Dong et al., 2019; 2020; Hubara et al., 2021; Li et al., 2021; Yao et al., 2021), including via adaptive methods (Nagel et al., 2020). SmoothQuant (Xiao et al., 2023) rescales between activations and weights to remove outliers from the activations and make quantization overall easier. ZeroQuant (Yao et al., 2022) proposes a per-layer knowledge distillation method. LLM.int8() (Dettmers et al., 2022) decompose matrix multiplications into a majority of 8 bit and a minority of 16 bit operations. LUT-GEMM (Park et al., 2023) designs kernels to accelerate quantized matrix multiplications. RPTQ (Yuan et al., 2023) reorders activations and quantizes them in groups, reducing the impact of range differences between channels.

**Running LLMs on Consumer GPUs**   Our methods for 3-bit and 4-bit precision enable the finetuning of a 65B LLM on a 48GB GPU, and a 30B LLM on a 24GB GPU. Additionally, our 2-bit approach allows for the finetuning of a 65B LLM on a 24GB GPU, making the finetuning of LLMs accessible on consumer hardware. Moreover, fitting an entire LLM on GPU unlocks data parallelism, which is more efficient than model parallelism. Previous 8-bit quantization methods required a 96GB GPU to fully fit a 65B model. Finetuning GPUs on consumer hardware holds promise to accelerate model iteration and apply LLMs to a wider range of domains by a larger number of practitioners.

## 5.3   What is a Good Base LLM for Finetuning?

The traditional measure of a base LLM is perplexity. In the adjacent table, we report LLaMA perplexity (PPL) on Wiki2 as well as finetuning performance on BBH. Interestingly, the correlation is not perfect: large gaps in PPL admit small gaps in BBH. This questions LLM evaluation when the goal is finetuning, and suggests exploring new training strategies.

| Models | Quantization | BBH | PPL |
|--------|------------|------|------|
| LLaMA (13B) | 3-bit | 35.3 | 6.63 |
| | 4-bit | 36.2 | 5.36 |
| LLaMA (65B) | 3-bit | 43.3 | 5.04 |
| | 4-bit | 43.7 | 3.84 |

Table 8: BBH vs. PPL

More generally, our results provide empirical evidence that high performance on downstream tasks can be achieved with a smaller quantized LLM than previously thought. While existing methods (e.g., LLM.int8()+LoRA; Dettmers et al. (2022)) operate in 8 bits, we find that 2-bit, 3-bit, or 4-bit finetuning yields the best results for a fixed bit budget. For example, we find that 4-bit and 3-bit 65B models outperform 8-bit and 16-bit 30B models on instruction following tasks. On the SAMSum summarization task, we find that 3-bit models are able to attain a new state-of-the-art ROUGE score, and 2-bit models match the performance of 8-bit models quantized with LLM.int8(). The high performance of these low-precision models suggests that competitive finetuning performance can be achieved on any base quantized LLM with x-bit precision, provided that the LLM exhibits reasonably good performance from the beginning.

## 5.4   Limitations

An advantage of LoRA is that it has low inference overhead, since the low-rank adaptor can be added in to the full-precision weight matrix when deploying. One limitation of MODULoRA is that it does not share this advantage relative to the black-box quantized model: the low-rank adaptor cannot be trivially added to the weight matrix because the weight matrix is quantized while the adaptor is not. So, the weight matrix

and adaptor cannot be fused readily, and an implementation as in Figure 1 is required at inference time. A second limitation of MODULORA is that making finetuning possible on widely available commodity hardware may make finetuning too easy, presenting potential problems related to LLM safety. Another limitation of MODULORA is that the largest models in use today (e.g. GPT-4) can have up to 1 trillion parameters, and even at the minimum of 1 bit per parameter this still would take up 125 GB, which exceeds memory on commodity GPUs: thus a straightforward application of MODULORA will be unable to make these largest-scale models finetunable on commodity hardware.

## 6    Conclusion

Finetuning large language models typically requires substantial hardware and storage resources. Our method, MODULORA, enables 2-bit finetuning of 65B models on a single 24GB consumer GPU and also supports 3-bit and 4-bit finetuning of the same models using a single 48GB GPU. At the core of our approach is a simple, quantization-agnostic backward pass that enables integrating low-rank adapters with frozen LLM weights obtained from a user-defined quantization module. By integrating with modern quantizers, MODULORA achieves state-of-the-art performance compared to both parameter-efficient and full fine-tuning techniques.

MODULORA's flexibility and competitive performance make finetuning more accessible and cost-effective in a resource-constrained setting. This assists open-source model development and facilitates scientific research. More broadly, we believe that MODULORA will help democratize access to large language models and make them available to a broader audience.

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

## A  Additional Implementation Details

### A.1  Configurations for BBH Evaluation

We evaluate the BBH dataset using LoRA adapter weights from huggingface hub with different configurations. For the Bits&Bytes 8-bit (LLM.int8()) LoRA adapter weights, we utilized two sources: the Alpaca-7B one is obtained from the 'tloen/alpaca-lora-7b' repository, while the weights for Alpaca-13b and 30b were sourced from 'chansung/alpaca-lora-xxb'. In the case of Bits&Bytes 4-bit (QLoRA) adapter weights, all configurations (Alpaca-7B, 13B, and 30B)—were uniformly accessed from 'timdettmers/qlora-alpaca-xxb'. Note that for the Bits&Bytes 4-bit (QLoRA) and Bits&Bytes 8-bit (LLM.int8()) adapter wights of the 65B model, we obtain them by finetuning the base 65B LLaMa model on Alpaca dataset using the same set of hyperparameters as ours.

## B  Additional Empirical Experiments

### B.1  Additional Experiments on Code-Alpaca with LLaMA

We conducted additional experiment on Code-Alpaca ((Chaudhary, 2023)). The result is shown in Table 9. Consistent with our hypothesis, MODULoRA performs better than or at least on par with the higher precision 8-bit models given the same number of trainable parameters and set up.

| Code Alpaca Performance | 7B | 13B | 30B | 65B |
|---|---|---|---|---|
| LLMTOOLS (3-bit) | 53.6 / 36.3 / 50.7 | 57.0 / 40.0 / 53.3 | 58.1 / 40.7 / 54.3 | 60.0 / 44.1 / 58.8 |
| LLMTOOLS (4-bit) | 54.6 / 37.2 / 51.4 | 57.4 / 40.6 / 54.3 | 59.0 / 41.4 / 57.5 | 60.2 / 43.5 / 56.8 |
| Bits&Bytes 8-bit (LLM.int8()) | 54.0 / 36.3 / 50.9 | 57.7 / 41.3 / 54.9 | 60.6 / 43.5 / 57.5 | 61.1 / 44.1 / 58.0 |

Table 9: Instruction-tuned models evaluated using ROUGE 1/2/L-Sum on Code Alpaca in 3, 4, and 8 bits.

### B.2  Finetuning & Inference Latency

We conducted experiment to test the finetuning and inference latency of MODULoRA.

**Finetuning**. During finetuning, MODULoRA significantly outperforms full-precision LoRA as show in table 10, reducing the training time by approximately 59.3% and memory usage by 91.5%. This efficiency in finetuning speed is primarily attributed to reduced data movement within the GPU memory.

**Inference**. During inference, MODULoRA has a slightly lower speed compared to LoRA and QLoRA as shown in table 11. We attribute this to the use of CUDA kernels that are currently not as optimized as those of QLoRA. Note that

| Precision | LLMTools (2-bit) | QLoRA (4-bit) | LoRA (Full Precision) |
|---|---|---|---|
| Seconds/Iteration | 0.61 s/it | 0.80 s/it | 1.50 s/it |

Table 10: Finetuning speed for LLAMA 7B on MNLI-m benchmark with batch size 1. We report the average time to complete one step for one training data entry. To ensure fair comparison, we use a single A6000 to run on all three methods.

| Precision | LLMTools (2-bit) | QLoRA (4-bit) | LoRA (Full Precision) |
|---|---|---|---|
| Seconds/Iteration | 0.68 s/it | 0.52 s/it | 0.52 s/it |

Table 11: Inference speed for LLAMA 7B on MNLI-m benchmark. We report the average time to complete inference for one evaluation data entry. To ensure fair comparison, we use a single A6000 to run on all three methods.

# C  Hyperparamters Used in Experiments

## C.1  LLaMA / OPT on SAMSum

We set up the training procedure following Hu et al. (2022), with particular accommodation to our particular language models. For a fair comparison with the concurrent work QLoRA, we use the exact same hyperparameter set up as shown in Table 12 . We train using AdamW for 350 steps with a batch size of 128 samples. We report the results over 3 random seeds; the result for each run is taken from the training steps with the lowest validation loss.

| Dataset | Model | LLaMA 7B / 13B / 30B / 65B    OPT 7B/ 13B / 30B |
|---------|-------|-------------------------------------------------|
| | Optimizer | AdamW |
| | Warmup Ratio | 0.06 |
| SAMSum | Batch size | 128 |
| | Evaluation Batch size | 16 |
| | Evaluation Steps | 50 |
| | Total # Training Steps | 350 |
| | Learning Rate Schedule | Cosine |
| | Learning Rate | 1e-3 |
| | WeightDecay | 0.0 |
| | LoRAConfig | $r_q = r_v = 8$ |
| | LoRA $\alpha$ | 32 |
| | Max Seq. Len | 250 |

Table 12: Hyperparamters configuration for ModuLoRA, Q-LoRA on SAMSum

## C.2  LLaMA on Code-Alpaca & Text-Classification

We again train using AdamW optimizer with a warmup ratio of 0.06. We tune learning rate, batch size, training steps for each task. We report the results over 3 random seeds. The result for each run is taken from the training steps that yield the lowest validation loss.

| Dataset | LLaMA Model | 13/30/65 B | Dataset | LLaMA Model | 7/13/30/65 B |
|---------|-------------|------------|---------|-------------|--------------|
| | Optimizer | AdamW | | Optimizer | AdamW |
| | Warmup Ratio | 0.06 | | Warmup Ratio | 0.06 |
| Text-Classification | Batch size | 256 | Code-Alpaca | Batch size | 128 |
| | Evaluation Batch size | 32 | | Evaluation Batch size | 4 |
| | Evaluation Steps | 100 | | Evaluation Steps | 40 |
| | Total # Training Steps | 1000 | | Total # Training Steps | 120 |
| | Learning Rate Schedule | Cosine | | Learning Rate Schedule | Linear |
| | Learning Rate | 1e-3 | | Learning Rate | 1e-3 |
| | WeightDecay | 0.0 | | WeightDecay | 0.0 |
| | LoRAConfig | $r_q = r_v = 8$ | | LoRAConfig | $r_q = r_v = 8$ |
| | LoRA $\alpha$ | 32 | | LoRA $\alpha$ | 32 |
| | Max Seq. Len | 128 | | Max Seq. Len | 165 |

Table 13: Hyperparamters configuration for ModuLoRA, Q-LoRA on Text-Classification

Table 14: Hyperparamters configuration for ModuLoRA, Q-LoRA on Alpaca-Code

### C.3 LLaMA on MNLI-M

Training is conducted using the AdamW optimizer, with a warmup ratio set at 0.06. We tune the learning rate, batch size, and training steps. Results are reported over three random seeds, and for each run, the performance metric is derived from the training step with the lowest validation loss. See Table 15 for more details on the hyperparameters used.

| Dataset | Model | LLaMA 7B / 13B / 30B / 65B |
|---------|-------|----------------------------|
|         | Optimizer | AdamW |
|         | Warmup Ratio | 0.06 |
| MNLI-M  | Batch size | 128 |
|         | Evaluation Batch size | 64 |
|         | Evaluation Steps | 64 |
|         | Total # Training Epoch | 1.0 |
|         | Learning Rate Schedule | Cosine |
|         | Learning Rate | 1e-3 |
|         | WeightDecay | 0.0 |
|         | LoRAConfig | $r_q = r_v = 8$ |
|         | LoRA $\alpha$ | 32 |
|         | Max Seq. Len | 128 |

Table 15: Hyperparamters configuration for ModuLoRA, Q-LoRA on MNLI-M

### C.4 LLaMA on Alpaca for BBH Evaluation

Training is conducted using the AdamW optimizer, with a warmup ratio set at 0.06. We tune the learning rate, batch size, and training steps. Results are reported over three random seeds. See Table 16 for more details on the hyperparameters used.

| Dataset | Model | LLaMA 7B / 13B / 30B / 65B |
|---------|-------|----------------------------|
|         | Optimizer | AdamW |
|         | Warmup Ratio | 0.06 |
| Alpaca  | Batch size | 128 |
|         | Total # Training Epochs | 3 |
|         | Learning Rate Schedule | Linear |
|         | Learning Rate | 1e-3 |
|         | WeightDecay | 0.0 |
|         | LoRAConfig | $r_q = r_v = 8$ |
|         | LoRA $\alpha$ | 16 |
|         | Max Seq. Len | 256 |

Table 16: Hyperparamters configuration for ModuLoRA on Alpaca

