# OpenReview forum: "ModuLoRA: Finetuning 2-Bit LLMs on Consumer GPUs by Integrating with Modular Quantizers"
_TMLR — Accepted by TMLR_

### Review · Reviewer_eLwS · 2023-10-09

**Summary Of Contributions:**

This work introduces a technique that combines weight quantization with the LoRA parameter efficient finetuning strategy (ModuLoRA). The main target is to train LLM in memory-constrained hardware. ModuLoRA is quantizer-agnostic. Dequantized weights are not stored but recalculated during the backward pass to preserve the memory savings of the few-bit representation. ModuLoRA enables LLM finetuning at 3 and 4 bits, resulting in performance comparable to baseline. Results are presented primarily on the LLAMA model family, on several downstream tasks. The authors are sharing this work as part of a library (LLMTools).

**Audience:**

Yes

**Broader Impact Concerns:**

Some broader impact concerns are briefly addressed in the manuscript. No additional concern on my end.

**Claims And Evidence:**

Yes

**Requested Changes:**

1. Discuss ModuLoRA overheads for finetuning and inference

2. List and compare actual memory requirements of the presented models

3. Rewrite, expand, or remove appendices A.1 and A.2

**Strengths And Weaknesses:**

Strengths:

- The topic of this work is of clear interest for ML practitioners

- The paper is well structured, well written, very easy to follow

- Solid results on various downstream tasks with models finetuned down to 3 bits: often comparable to their 8-bit counterpart, or showing limited degradation

- ModuLoRA allows for a flexible choice of quantizers, leaving the path open for further improvements as new quantizers are developed

- The authors present direct comparisons against the recently proposed QLoRA, arguably the low-precision finetuning method conceptually closest to ModuLoRA, and discuss differentiating factors. ModuLoRA results compare favorably with QLoRA

- The shared library is a very valid tool to democratize LLM finetuning

Weaknesses:

- Quantization is applied solely to weights, therefore while achieving the goal of finetuning in memory-constrained environments, computational speed ups are not to be expected. This is partly acknowledged in the paper but it is unclear what is the extent of ModuLoRA's overheads. I would expect they may also depend on the choice of quantizer. How does finetuning runtime compares with LoRA and QLoRA? What about inference latency / throughput?

- The paper mentions that in some scenario ModuLoRA materializes a memory advantage against QLoRA, however no evidence is provided to support this statement. In fact, there is no concrete mention of memory requirements for various models / quantization / batch size combinations in the paper, aside from some upper bounds (such as one of the main claims of being able to finetune a 65B model on a single 48G GPU). As this relates to the main takeaway of this paper, I recommend including a table comparing memory requirements, at the very least as an appendix.

- Appendices A.1 and A.2 do not appear to add any information to the main text and should be either expanded or removed

---

> ### Author Response · Authors · 2023-12-05
> **Response to reviewer eLwS (1/1)**
>
> We thank the reviewer for their constructive feedback. We address their concerns below.
>
> **Concern 1: Discussing the overhead presented by ModuLoRA during finetuning and inference.**
>
> We provide the runtime and memory usage of ModuLoRA in the tables below.
>
> **Training Time.** If we eliminate weight re-materialization, all model parameters are held in memory in full precision, and our method reduces to standard full-precision LoRA. We observe that finetuning our 4-bit LLAMA-7B models on a single A6000 GPU results in a reduction in training time of approximately 30% compared to using full-precision LoRA and in an 87.5% reduction in memory usage. ModuLoRA achieves faster finetuning speed than LoRA due to reduced data movement within the GPU memory.
>
> | Precision         | LLMTools (2-bit) | QLORA (4-bit) | Full Precision LoRA |
> |-------------------|--------------|------------|---------------------|
> | **Time/Iteration**| 0.6s/it     | 0.8s/it    | 1.5s/it             |
> *Table 1: Finetuning speed for LLAMA 7B on MNLI-m benchmark with batch_size = 1. To ensure fair comparison, we use a single A6000 for all three methods.*
>
>
> | Precision         |  LLMTools (2-bit) | QLORA (4-bit) | Full Precision LoRA |
> |-------------------|--------------|------------|---------------------|
> | **Memory Footprint**        |  3.2 GB  | 5.2 GB     | 38.4 GB             |
> *Table 2: Finetuning footprint for LLAMA 7B on MNLI-m benchmark with batch_size = 1. To ensure fair comparison, we use a single A6000 for all three methods.*
>
> Thus, despite the added recomputation (rematerialization) process, ModuLoRA does not introduce a computational overhead relative to LoRA; of course, the cost of this speed-up is the need to use quantized LLM weights. On some tasks (e.g., BigBenchHard), quantization introduces small drops in accuracy, although on other tasks (e.g., SamSum, MNLI), it does not.
>
> While ModuLoRA is 20% slower than QLoRA, our current implementation uses relatively simple CUDA code. We may close this performance gap with further CUDA optimizations or by integrating with 2-bit quantizers (which require even less memory transfer).
>
> **Inference Time.** During inference, ModuLoRA has a slightly lower speed compared to LoRA and QLoRA. We again attribute this to the use of CUDA kernels that are currently not as optimized as those of QLoRA. Note that we can achieve the speed of full-precision LoRA by materializing all of the weights at the cost of higher memory usage.
>
>
> | Precision         | LLMTools (2-bit) | QLORA (4-bit) | Full precision LoRA |
> |-------------------|--------------|------------|---------------------|
> | **Seconds/Iteration**  | 0.68 it/s      | 0.52 it/s   | 0.52 it/s      |
>
> *Table 3: Inference speed for LLAMA 7B on MNLI-m benchmark. We report the average time to complete inference for one evaluation data entry. To ensure fair comparison, we use a single A6000 for all three methods.*
>
> **Concern 2: Listing and comparing actual memory requirements of the presented models.**
>
> We present a quantitative analysis of the memory costs associated with ModuLoRA, QLoRA, and LoRA. The accompanying table, which is also included in the appendix, outlines the specific GPU memory requirements for each experiment. Notably, ModuLoRA demonstrates a substantial reduction in memory usage, **saving up to approximately 93% compared to full-precision LoRA** during the finetuning of LLAMA models.
>
> | Model Precision  | 7b    | 13b   | 33b    | 65b    |
> |-----------------|-------|-------|--------|--------|
> |  LLMTools (2-bit)  | 3GB  | 5GB | 11GB | 22GB |
> |  QLORA (4-bit) | 5GB | 9GB | 20GB | 37GB |
> | Full precision  | 38GB  | 74GB| 183GB| 360GB |
> *Table 4: GPU Memory Requirement of finetuning LLAMA model sets on MNLI-M dataset with batch_size = 1.*
>
> **Concern 3: The need to rewrite, expand, or remove appendices A.1 and A.2.**
>
> Thank you for this feedback. We have removed A.1 and A.2 from the Appendix as you suggested.

---

### Review · Reviewer_E1B8 · 2023-10-16

**Summary Of Contributions:**

Summary of contributions
1. The paper studies the problem of the quantization aware fine-tuning of the large language models, specifically dealing with the Low-rank adapter based fine-tuning of the models.
2. The core contribution is proposing a recompilation methods in which the based weight matrix in LoRA is computed twice, one during the inference, one during the back-propagation. In this way, the full precision (or fp16) version of the based weight matrix can be freed after a layer and is not needed to be stored in GPU memory.
3. The paper also releases an open-source tools called LLMTools for the quantization of LLMs and release several quantized popular models.

**Audience:**

Yes

**Broader Impact Concerns:**

no broader impact concerns

**Claims And Evidence:**

Yes

**Requested Changes:**

1. More insights on why the 4bits model of 13B Lamma and 30B lamma perform worse than 3bits.
2. Since the number of training epochs/tokens are important for performance, the evaluation sections should provide more details on the training settings. Especially, the fine-tuning setting when comparing to QLoRA should be the same for fairness.
3. It would be good to provide more details on the speed of finetuning, and show the trade off between the recomputation and longer training time.
4. Quantitative results of the memory saving of the proposed method.

**Strengths And Weaknesses:**

Strength
1. The proposed method is simple and effective to reduce GPU memory consumption for fine-tuning quantized LLM models.
2. It is agnostic to the specific quantization method.
3. The evaluation results shows convincing results that the 3bit model can provide similar accuracy to a fp16 model.

Weaknesses:
1. The recompilation introduces additional computational overhead.
2. More direct comparisons of the memory usage during training should be provides.

---

> ### Author Response · Authors · 2023-12-05
> **Response to reviewer E1B8 (1/2)**
>
> We thank the reviewer for their constructive feedback. We address their concerns below.
>
> **Concern 1: Why did some 4-bit models perform worse than 3-bit models on the text classification task?**
>
> Our analysis indicates that the 4-bit models performed worse than 3-bit models on some tasks because of overfitting: on some tasks, **the stronger quantization acted as a form of regularization that improved performance by reducing overfitting.**
>
> Below, we report the training / evaluation loss and test classification accuracy of the 13B and 30B model. We observe that the gap between train loss and eval loss in 4-bit models is larger than the gap in 3-bit models. This evidence suggests that 4-bit models tend to overfit on the simple tasks, which indicates that stronger quantization could regularize the model and improve its performance by preventing overfitting.
>
> | Precision		| 13B  			| 30B  		|
> |------|------|------|
> | LLMTools (3-bit)	|  1.6727 / 1.6751 	|1.6243	/ 1.6312	|
> | LLMTools (4-bit)	| 1.6588 / 1.6682 	|1.6157 / 1.6254	|
> *Table 1: Text classification training/evaluation loss for 13B, 30B LLAMAs finetuned with LoRA & ModuLoRA in 3, 4 bits.*
>
> | Precision    | 13B  | 30B   |
> |---|---|----|
> | LLMTTools (3-bit)         | 93.5 ± 0.7   | 97.0 ± 0.9   |
> | LLMTTools (4-bit)         | 92.9 ± 0.7   | 96.3 ± 1.0   |
> *Table 2: Text classification accuracy (%) for 13B, 30B LLAMAs finetuned with ModuLoRA in 3, 4 bits.*
>
> Note that we observe this gap in performance on the easiest task (text classification), but not on more challenging tasks (text summarization, instruction following, etc). Similar observations have been noted in the GPTQ paper [1], where the 560M and 1.1B 4-bit BLOOM models performed worse than 3-bit models on ARC-easy but showed improved performance on ARC-challenge. We include the OPTQ results table here:
>
> | BLOOM | Bits | 560M  | 1.1B  | 1.7B  |
> |---|---|---|---|---|
> | full  | 16   | 41.71 | 45.41 | 48.11 |
> | RTN   | 4    | 39.40 | 42.51 | **44.70**|
> | GPTQ  | 4    | **40.24** | **44.49** | 44.49 |
> | RTN   | 3    | **45.44** | **46.87** | 37.58 |
> | GPTQ  | 3    | 39.14 | 41.79 | **42.85** |
>
> *Table 3: GPTQ Bloom Accuracy on ARC-easy [1]. Note that the 3-bit RTN outperforms the advanced quantization algorithm OPTQ at 4-bit precision, particularly in smaller models (560M and 1.1B)*
>
>
> | BLOOM | Bits | 560M    | 1.1B    | 1.7B    |
> |---|----|---|-----|-----|
> | full  | 16   | 24.15   | 25.68   | 26.79   |
> | RTN   | 4    | **23.89** | 23.34   | **26.45** |
> | GPTQ  | 4    | 23.46   | **25.51** | 25.94   |
> | RTN   | 3    | 21.67   | 22.86   | 23.29   |
> | GPTQ  | 3    | **23.21** | **24.06** | **24.91** |
> *Table 4: GPTQ Bloom Accuracy on ARC-Challenge [1]. In more challenging tasks, 4-bit OPTQ outperforms the 3-bit RTN for all model sizes.*
> [1] Frantar, Elias, et al. 'GPTQ: Accurate Post-Training Quantization for Generative Pre-trained Transformers.' arXiv preprint arXiv:2210.17323 (2022).
>
> **Concern 2:  The evaluation sections should provide more details on the training settings (e.g., number of training steps), especially, the fine-tuning setting when comparing to QLoRA.**
>
> Please note that **we maintained identical configurations and hyperparameter settings for finetuning using both QLoRA and ModuLoRA.** Detailed information about these hyperparameters can be found in Appendix D. We have moved additional details to the evaluation section of the main paper, as requested by the reviewer.
>
> In addition, the fairness and accuracy of our comparison are evident from the evaluation code scripts of both QLoRA and ModuLoRA, which we will link here (please refer to line #94-106 in both scripts, where we set up our hyperparameters).
>
> [LLMTools Training scripts for SAMSum](https://anonymous.4open.science/r/MODULoRA-Experiment-TMLR/finetune/samsum-llama/train_samsum_4bit.py)
>
> [QLoRA Training scripts for SAMSum](https://anonymous.4open.science/r/MODULoRA-Experiment-TMLR/finetune/samsum-llama/train_samsum_4bit_bnb.py)

---

> ### Author Response · Authors · 2023-12-05
> **Response to reviewer E1B8 (2/2)**
>
> **Concern 3:  Providing more details on the speed of finetuning and showing the trade off between the recomputation (of the quantized weights) and longer training time.**
>
> We provide the runtime of finetuning in the tables below. Note that if we eliminate recomputation, all model weights are held in memory in full precision, and our method reduces to standard full-precision LoRA. **We observe that finetuning 2-bit LLAMA-7B models on a single A6000 GPU results in a reduction of  training time by approximately 60% compared to using full-precision LoRA and in an 87% reduction in memory usage.** ModuLoRA achieves faster finetuning speed than LoRA due to reduced data movement within the GPU memory.
>
> Despite the added recomputation (rematerialization) process, ModuLoRA does not introduce a computational overhead relative to LoRA; of course, the cost of this speed-up is the need to use quantized LLM weights. On some tasks (e.g., BigBenchHard), quantization introduces a drop in accuracy, although on other tasks (e.g., SamSum, MNLI), it does not.
>
> | Precision         | LLMTools (2-bit) | QLORA (4-bit) | Full Precision LoRA |
> |-------------------|--------------|------------|---------------------|
> | **Time/Iteration**| 0.6s/it     | 0.8s/it    | 1.5s/it             |
> *Table 5: Finetuning speed for LLAMA 7B on MNLI-m benchmark with batch_size = 1. To ensure fair comparison, we use a single A6000 for all three methods.*
>
>
> | Precision         |  LLMTools (2-bit) | QLORA (4-bit) | Full Precision LoRA |
> |--|--|---|--|
> | **Memory Footprint**        | 3.2 GB       | 5.2 GB     | 38.4 GB             |
> *Table 6: Finetuning footprint for LLAMA 7B on MNLI-m benchmark with batch_size = 1. To ensure fair comparison, we use a single A6000 for all three methods.*
>
> **Concern 4: Reporting quantitative results on the memory savings of the proposed method.**
>
> In the table below, we present a detailed quantitative analysis of the memory savings achieved by ModuLoRA in comparison to LoRA. Our findings indicate that ModuLoRA can **reduce memory costs by up to approximately 93% compared to full-precision LoRA** when finetuning LLAMA models. This significant reduction highlights the efficiency of ModuLoRA in optimizing memory usage.
>
> (Please consider taking a look at the newly added Section 4.6 “Memory Requirements”, where we provide additional details and graphs.)
>
> | Model Precision  | 7b    | 13b   | 33b    | 65b    |
> |--|--|--|--|--|
> |  LLMTools (2-bit)  | 3GB  | 5GB | 11GB | 22GB |
> |  QLORA (4-bit) | 5GB | 9GB | 20GB | 37GB |
> | Full precision  | 38GB  | 74GB| 183GB| 360GB |
> *Table 7: GPU Memory Requirement of finetuning LLAMA model sets on MNLI-M dataset with batch_size = 1.*

---

### Review · Reviewer_RBFJ · 2023-11-21

**Summary Of Contributions:**

This paper introduces an efficient framework to finetune large language models with less memory. Specifically, 3-bit OPTQ quantizer is utilized to all linear layers and then the weight quantizer is incorporated into low-rank adapters to enable a memory-efficient fine-tuning. The authors also provide a library for this framework with efficient CUDA implementations. The experiments on various LLM tasks show the effectiveness of the proposed algorithm.

**Audience:**

Yes

**Broader Impact Concerns:**

None.

**Claims And Evidence:**

Yes

**Requested Changes:**

1. Please justify the novelty in case I miss some contributions.
2. Please provide more evaluation of latency.
3. Please examine and revise the typos.

**Strengths And Weaknesses:**

Strength:
1. The paper is easy to follow.
2. The provided library could contribute to efficient LLM community.
3. The experiments include various tasks and the results seem promising.

Weakness:
1. The novelty is limited. The main contribution of this work lies in the framework of fine-tuning with low precision layers, which can be treated as a simple combination of LORA and OPTQ.
2. Lack of evaluation of latency in the experiments.
3. There exist many typos in the paper and the authors are suggested to check them carefully. Part of minors are listed below:

*  In abstract, natural language infernece -> natural language inference
* In Section 2.2, The method iteretiavely runs -> The method iteratively runs
* In Section 3.1, quantized weighs -> quantized weights

---

> ### Author Response · Authors · 2023-12-05
> **Response to reviewer RBFJ (1/2)**
>
> We thank the reviewer for their constructive feedback. We address their concerns below.
>
> **Concern 1: Explaining the novelty of the method.**
>
> We argue that ModuLoRA has substantial novelty and is more than just the combination of LoRA and OPTQ. The novelty can be summarized as follows:
>
> - **Modularity with diverse quantizers**: ModuLoRA integrates with any weight quantizer, unlike QLoRA, which uses a specific quantization scheme. This flexibility allows quantizing using even advanced 2-bit quantization techniques like QUIP [2].
> - **Low-precision finetuning**: ModuLoRA is the first method to enable 3-bit LLM finetuning, with an upcoming integration with QUIP that supports 2-bit finetuning, surpassing QLoRA's 4-bit limit.
> - **Conceptual Simplicity**: The core technical idea of ModuLoRA is adaptive weight re-materialization. This idea is simple, yet also very general (i.e., it works with any quantizer), and it is surprising that it yields extremely performant finetuning algorithms.
> - **Memory efficiency**: ModuLoRA allows finetuning of large models on consumer-grade GPUs. The release of LLMT00LS, featuring ModuLoRA, democratizes access to LLM finetuning.
> - **Proven Performance**: We provide convincing empirical evidence showing ModuLoRA's effectiveness in achieving high performance on downstream tasks with smaller LLMs.
>
> [2] Chee, Jerry, et al. "QUIP: 2-Bit Quantization of Large Language Models With Guarantees." arXiv preprint arXiv:2307.13304 (2023).

---

> ### Author Response · Authors · 2023-12-05
> **Response to reviewer RBFJ (2/2)**
>
> **Concern 2: Reporting additional results regarding finetuning and inference latency.**
>
> We provide the runtime and memory usage of ModuLoRA in the tables below.
>
> **Training Time.** If we eliminate weight re-materialization, all model parameters are held in memory in full precision, and our method reduces to standard full-precision LoRA. We observe that finetuning 2-bit LLAMA-7B models on a single A6000 GPU results in a **reduction of training time by approximately 60% compared to using full-precision LoRA and in an 87% reduction in memory usage**. ModuLoRA achieves faster finetuning speed that LoRA due to reduced data movement within the GPU memory.
>
> | Precision         | LLMTools (2-bit) | QLORA (4-bit) | Full Precision LoRA |
> |-------------------|--------------|------------|---------------------|
> | **Time/Iteration**| 0.6s/it     | 0.8s/it    | 1.5s/it             |
> *Table 1: Finetuning speed for LLAMA 7B on MNLI-m benchmark with batch_size = 1. To ensure fair comparison, we use a single A6000 for all three methods.*
>
> | Precision         |  LLMTools (2-bit) | QLORA (4-bit) | Full Precision LoRA |
> |--|--|---|--|
> | **Memory Footprint**        | 3.2 GB       | 5.2 GB     | 38.4 GB             |
> *Table 2: Finetuning memory footprint for LLAMA 7B on MNLI-m benchmark with batch_size = 1. To ensure fair comparison, we use a single A6000 for all three methods.*
>
> Thus, despite the added recomputation (rematerialization) process, ModuLoRA does not introduce a computational overhead relative to LoRA; of course, the cost of this speed-up is the need to use quantized LLM weights. On some tasks (e.g., BigBenchHard), quantization introduces small drops in accuracy, although on other tasks (e.g., SamSum, MNLI), it does not.
>
> **Inference Time.** During inference, ModuLoRA has a slightly lower speed compared to LoRA and QLoRA. We attribute this to the use of CUDA kernels that are currently not as optimized as those of QLoRA. Note that we can achieve the speed of full-precision LoRA by materializing all of the weights at the cost of higher memory usage.
>
> | Precision         | LLMTools (2-bit) | QLORA (4-bit) | Full precision LoRA |
> |-------------------|--------------|------------|---------------------|
> | **Seconds/Iteration**  | 0.68 it/s      | 0.52 it/s   | 0.52 it/s     |
> *Table 3: Inference speed for LLAMA 7B on MNLI-m benchmark. We report the average time to complete inference for one evaluation data entry. To ensure fair comparison, we use a single A6000 for all three methods.*
>
> **Concern 3: Listing and comparing actual memory requirements of the presented models.**
>
> We present the detailed quantitative analysis of the memory costs associated with ModuLoRA, QLoRA, and LoRA. The accompanying table, which is also included in the appendix, outlines the specific GPU memory requirements for each experiment. Notably, ModuLoRA demonstrates a substantial reduction in memory usage, saving up to approximately 93% compared to full-precision LoRA during the finetuning of LLAMA models.
>
> (Please consider taking a look at the newly added Section 4.6 “Memory Requirement”, where we provide details and graphs that shows our minimal memory footprint.)
>
> | Model Precision  | 7b    | 13b   | 33b    | 65b    |
> |-----------------|-------|-------|--------|--------|
> |  LLMTools (2-bit)  | 3GB  | 5GB | 11GB | 22GB |
> |  QLORA (4-bit) | 5GB | 9GB | 20GB | 37GB |
> | Full precision  | 38GB  | 74GB| 183GB| 360GB |
> *Table 4: GPU Memory Requirement of finetuning LLAMA model sets on MNLI-M dataset with batch_size = 1*
>
>
> **Additional Comments**
>
> We also thank the reviewer for identifying typos in our work. We have fixed those typos as you suggested.

---

### Author Response · Authors · 2023-12-05
**Response to all reviewers**

We are grateful to all reviewers for their constructive feedback. In response to their concerns, we have made the following enhancements to our manuscript.

**Additional experimental evaluation.** We introduce Section 4.6, titled 'Memory Requirements,' in which we evaluate the memory usage of ModuLoRA (see Table 7 and Figure 2 in our revised manuscript). We also provide additional benchmarks on speed in Appendix B.2, 'Finetuning & Inference Latency.'

**Improved comparison to concurrent work.** We provide additional information on how we compare our method ModuLoRA against QLoRA. We confirm that both methods use an identical experimental setup.

**Additional insights/clarifications.** We explain certain discrepancies  between 3-bit and 4-bit performance on the MNLI task. We also provide additional arguments in support of the novelty of our method.

**Integration with 2-bit quantizers.** We have recently integrated 2-bit QuIP quantization with ModuLoRA. We hope to share later this week additional results which will further strengthen our paper.

We appreciate the opportunity to improve our manuscripts based on reviewer’s feedback.

---

### Author Response · Authors · 2023-12-17
**Response to All Reviewers: Update on 2-Bit QuIP# Quantization Integration**

We are writing to share some exciting updates regarding our latest integration of 2-bit QuIP# quantization, which enables finetuning a 65B LLMs on a single 24GB GPU for the first time. On a standard summarization task, we find that **2-bit ModuLoRA+QuIP# outperforms 4-bit QLoRA** in terms of ROUGE-L on large model sizes.

Following OPTQ, Chee et al. proposed QuIP, a quantization algorithm that makes two-bit LLM compression viable for the first time [1]. The method follows a 2-step procedure: (1) an adaptive rounding procedure that minimizes a quadratic proxy objective, (2) an efficient pre- and post-processing procedure ensuring weight and Hessian incoherence through multiplication by random orthogonal matrices. Further, Tseng et al. proposed QuIP#, which combines lattice codebooks with incoherence processing from QuIP to create state-of-the-art 2 bit quantized models [2]. We show the performance of QuIP# (with $D_4$ codebooks) quantized LLMs on the SAMSum summarization experiment.

We finetune 7B-65B LLAMA1 models in 2-bit precision on the SAMSum summarization task. ModuLoRA models (2-bit) match the performance of 8-bit LoRAs in BitsAndBytes and LLM.int8() and 4-bit LoRAs in BitsAndBytes and QLoRA. Switching from 4-bit to 3-bit, and then from 3-bit to 2-bit, precision within ModuLoRA reduces ROUGE by only about 1%.

| Baselines      | Models | Finetuning Adaptation | # Trainable Parameters | SAMSsum (Rouge 1/2/L) |
|----------------|--------|-----------------------|------------------------|-----------------------|
|                | GPT-3  | Full Finetuning       | 175,255.8M             | 52.0 / 28.0 / 44.5    |
|                | GPT-3  | Adapter               | 40.1M                  | 53.2 / 29.0 / 45.1    |
|                | GPT-3  | LoRA                  | 4.7M                   | 53.8 / 29.8 / 45.9    |
|                | Pegasus| SiLC                  | 2B                     | 54.4 / 29.9 / 45.9    |
|                |        |                       |                        |                       |

| LLAMA Finetuning      | Quantizer | 7B | 13B | 30B |65B |
|----------------|--------|-----------------------|------------------------|-----------------------|--------------------|
| **LLMTools (2-bit)** | **QuIP# $(D_4)$**  | **42.7 / 49.2 / 26.9 **   | **44.4 / 50.7 / 28.6 **     | **46.4 / 51.6 / 30.2 **    | **46.8 / 52.3 / 30.5 ** |
| LLMTools (3-bit)| OPTQ  | 44.0 / 51.2 / 28.2    | 45.1 / 52.4 / 29.6      | 46.3 / 53.6 / 30.8     | 46.5 / 54.1 / 30.9  |
| LLMTools (4-bit)| OPTQ  | 44.4 / 51.7 / 28.3   | 46.1 / 53.2 / 30.2      | 46.9 / 53.9 / 31.2     | 49.0 / 55.9 / 32.7  |
| Bits&Bytes (4-bit)| QLoRA | 44.5 / 51.6 / 28.3  | 44.1 / 51.3 / 28.1      | 45.7 / 53.0 / 30.2     | 45.9  / 53.8 / 30.5 |
| Bits&Bytes (8-bit)| LLM.int8() | 44.5 / 51.9 / 28.1  | 43.6 / 51.3 / 28.2  | 44.1 / 50.8 / 28.4     | 46.3 / 53.9 / 30.4  |
*Table 1: Abstractive summarization on the SAMSsum dataset evaluated using ROUGE L/1/2. All metrics have ±0.5 confidence intervals.*

ModuLoRA significantly reduces the memory requirements for finetuning LLMs. The table below presents a detailed comparison of the memory requirements for fine-tuning LLMs using ModuLoRA, QLoRA, and LoRA.

| Model Precision  | 7b    | 13b   | 33b    | 65b    |
|-----------------|-------|-------|--------|--------|
|  LLMTools (2-bit)  | 3GB  | 5GB | 11GB | 22GB |
|  QLORA (4-bit) | 5GB | 9GB | 20GB | 37GB |
| Full precision  | 38GB  | 74GB| 183 GB| 360GB |
*Table 2: GPU Memory Requirement of finetuning LLAMA model sets on MNLI-M dataset with batch_size = 1 *
We have updated the paper throughout to reflect our integration of 2-bit QuIP# quantization (please see revised Title, Abstract, Introduction, and other sections).

[1] Chee, Jerry, et al. "QUIP: 2-Bit Quantization of Large Language Models With Guarantees." arXiv preprint arXiv:2307.13304 (2023).

[2] Tseng, Albert, et al. “QuIP#: QuIP with Lattice Codebooks.” https://cornell-relaxml.github.io/quip-sharp/ (2023).

---

### Decision · Action_Editor_s5Rs · 2023-12-30

**Recommendation:** Accept as is

**Comment:**

Following detailed rebuttals from the authors + the additional result on 2-bit quantization, the reviewers all lean toward accepting the paper.  A main point of discussion was the comparison with QLoRA in terms of performance and differentiation, but the reviewers were satisfied after the authors provided thorough additional datapoints, e.g. "In their rebuttal to mine as well as other reviewers' comments, the authors addressed my main concerns thoroughly, including in the manuscripts additional information on latency and memory requirements. The proposed ModuLoRA technique does shows some runtime overheads compared to its direct counterpart QLoRA (4 bit), but it brings more flexibility in the choice of quantizer and achieves better performance down to 3-bit precision.". Given the claims are well supported by the evidence + broad target audience of this paper, I recommend a solid accept.

**Audience:**

There are many individuals interested in knowing the findings of this paper, which include not only academic and industry communities working on quantization and efficient LLM finetuning, but also the large open-source communities interested in finetuning larger models on consumer GPUs. The additional result from the authors on 2-bit QuIP# quantization enabling finetuning a 65B LLM on a single 24GB GPU presents a step forward for democratizing finetuning access to more people, and justifies the broader impact of the paper beyond an academic contribution.

**Claims And Evidence:**

Claim: we propose MODULORA, a memory-efficient finetuning method that operates over low-precision weights obtained via a user-specified black-box quantization module.

Evidence: Supported throughout the paper + additional results presented during the rebuttal. Solid results using 2-bit QuIP# quantization and 3-bit OPTQ quantization as black-box quantization modules. Especially the newest result on 2-bit QuIP# quantization enabled finetuning a 65B LLMs on a single 24GB GPU for the first time.

Claim: we release LLMTOOLS, a user-friendly Python library that features an implementation of MODULORA and
that enables users to easily finetune the largest LLMs on consumer GPUs

Evidence: Supported. The code + reproduction are available: https://anonymous.4open.science/r/MODULoRA/README.md

Claim: we provide empirical evidence that high
performance on downstream tasks can be achieved with a smaller LLM than previously thought.

Evidence: Supported. Section 5.3 has a discussion.